# Clinical effectiveness of screening for age-related macular degeneration: A systematic review

Dalila Fernandes Gomes[1]*, Daniel da Silva Pereira Curado[2], Rosângela Maria Gomes[3,4], Betânia Ferreira Leite[5], Maíra Catharina Ramos[1], Everton Nunes da Silva[1,6]

**1** Graduate Program in Collective Health, University of Brasilia, Brasília, Federal District, Brazil, **2** Department of Management and Incorporation of Health Technologies, Ministry of Health, Brasilia, Federal District, Brazil, **3** Department of Social Pharmacy, College of Pharmacy, Federal University of Minas Gerais, Belo Horizonte, Minas Gerais, Brazil, **4** Faculty of Health Sciences, Department of Pharmacy, University of Brasilia, Brasilia, Brazil, **5** Department of Medicine, Paulista School of Medicine, Federal University of Sao Paulo, Sao Paulo, Brazil, **6** Faculty of Ceilandia, University of Brasilia, Brasilia, Federal District, Brazil

* dalilafg@gmail.com

**Data Availability Statement:** All relevant data are within the paper and its Supporting Information files.

## Abstract

### Introduction

Age-related macular degeneration (AMD) is an eye disease that occurs in patients over 50 years old. Early diagnosis enables timely treatment to stabilize disease progression. However, the fact that the disease is asymptomatic in its early stages can delay treatment until it progresses. As such, screening in specific contexts can be an early detection tool to reduce the clinical and social impact of the disease.

### Objective

Assess the effectiveness of screening methods for early detection of AMD in adults aged 50 years or older.

### Methods

A systematic review of comparative observational studies on AMD screening methods in those aged 50 years or older, compared with no screening or any other strategy. A literature search was conducted in the MEDLINE (via PubMed), Embase, Cochrane Library and Lilacs database.

### Results

A total of 5,290 studies were identified, three of which met the inclusion criteria and were selected for the systematic review. A total of 8,733 individuals (16,780 eyes) were included in the analysis. The screening methods assessed were based on optical coherence tomography (OCT) compared with color fundus photography, and OCT and telemedicine testing compared to a standard eye exam.

**Funding:** The authors received no specific funding for this work.

**Competing interests:** The authors have declared that no competing interests exist.

## Conclusion

The systematized data are limited and only suggest satisfactory performance in early screening of the population at risk of developing AMD. OCT and the telemedicine technique showed promising results in AMD screening. However, methodological problems were identified in the studies selected and the level of evidence was considered low.

## Introduction

Age-related macular degeneration (AMD) is a chronic progressive disease that affects patients over the age of 50 and causes damage to the central area of the retina, leading to reduced visual acuity and even blindness [1,2]. Clinically, the disease is classified into early, intermediate and advanced stages depending on drusen formation, retinal pigment changes, and the presence of geographic atrophy or choroidal neovascularization. In high-income countries, AMD is the leading cause of irreversible blindness in people over 50 years old [1] and the limitations resulting from multimorbidity associated with reduced quality of life tend to increase the disease burden.

The main risk factor for AMD is advanced age, as well as smoking and genetic factors [3]. The incidence of the disease is also expected to increase worldwide, in line with the rise in population aging rates [4]. The estimated global prevalence of AMD is 8.69% in 45 to 85-year-olds [5] with an increase of 196 to 288 million cases expected between 2020 and 2040, respectively [6]. As such, the demand for screening, intervention and postintervention monitoring continues to grow.

Early diagnosis enables adequate treatment and a better prognosis, but can be delayed for different reasons, such as the fact that the disease is asymptomatic in its early stages; compensatory mechanisms in the brain that make it difficult for patients to notice changes in vision in the initial stages; involvement of non-dominant eye and lack of awareness about the disease [7–9]. Timely diagnosis enables patients to undergo drug treatment through intravitreal injection with antiangiogenic agents in order to stabilize disease evolution or improve visual acuity. To that end, screening in specific contexts and public awareness of AMD can be early detection tools to reduce the clinical and social impact of the disease [8].

According to the World Health Organization (WHO), the purpose of screening is to identify people in an apparently healthy population who are at higher risk of a health problem or a condition, so that an early treatment or intervention can be offered [10,11]. In some cases, such as antenatal screening, the aim is to provide information about an increased risk or condition to help people make an informed decision about their care or treatment [10]. Currently, there are no medical treatments for early or intermediate AMD, only evidence that antioxidant vitamin and mineral supplementation can delay progression to the advanced stage and the loss of visual acuity in people with signs of the disease. However, the scientific evidence is limited and there are several unanswered questions regarding antioxidant vitamin and mineral supplementation in the prevention of AMD, including at what stage the protective effect may be important and potential interactions with genetic effects and other risk factors for the disease, such as smoking. Furthermore, the safety of antioxidant vitamin and mineral supplementation is controversial [12,13].

In 2018, the National Institute for Health and Care Excellence (NICE) guideline does not recommend using antioxidant and zinc supplements for AMD. According to this guideline, although the AREDS study showed some beneficial effects of combined antioxidant

supplementation in a subgroup of participants, the effects of the individual formula components on AMD progression were unclear and one of the ingredients (beta carotene) was associated with a possible risk of lung cancer among smokers. The guideline also highlights that although the AREDS research group introduced a new formulation that excluded beta carotene in the AREDS2 study, study design limitations related to secondary randomization and no placebo control mean that the effect of this formulation on AMD disease progression remains unknown. A well-conducted randomized trial could therefore provide new evidence on the benefits and risks of individual components of antioxidant supplements [14]. In this scenario, the present systematic review evaluated whether AMD screening could be considered effective in identifying patients with early clinical signs of the disease should treatments for these stages of AMD become available in the future, as well as those who have progressed to advanced AMD for early treatment prior to their self-presentation.

In light of the above, it is important to discuss and implement new public health strategies with a view to promoting healthy aging and reducing inequalities related to this process. This study aimed at conducting a systematic review of randomized controlled trials (RCTs) and comparative observational studies to assess the effectiveness of screening methods for early detection of AMD in adults aged 50 years or older. To that end, evidence on the effectiveness of AMD screening methods was systematized with the aim of contributing to the debate on these techniques.

## Methods

This is a systematic review of RCTs and comparative observational studies on AMD screening methods in those aged 50 years or older, compared with no screening or any other strategy in primary care, outpatient or hospital settings. The study was registered on the PROSPERO (International Prospective Register of Systematic Reviews) platform under CRD42022315907 [15] and compiled in accordance with the Preferred Reporting Items for Systematic Reviews and Meta-Analyses (PRISMA) checklist [16,17].

### Eligibility criteria

RCTs and comparative observational studies, including quasi-experimental, cohort, cross-sectional and case-control studies that assessed the effectiveness of AMD screening methods for those aged 50 years or older were considered eligible for inclusion. Excluded were investigations that evaluated participants with a confirmed diagnosis of AMD or any other ocular disease, such as glaucoma and diabetic retinopathy, since the main objective of this systematic review was to analyze the effectiveness of screening, considering the eligible population selected, screening method and the outcomes assessed.

Studies that presented any screening method for early detection of AMD were included. Given that there were no established detection methods for inclusion, biomicroscopy, angiography, ophthalmoscopy, retinography, optical coherence tomography (OCT) and telemedicine-based screening compared to no screening or any other screening method targeting the at-risk population (adults aged 50 years or older) were considered. However, studies that investigated artificial intelligence techniques as a screening method and those with no control group were excluded. There were no restrictions on language or date of publication.

Eligible studies were those that analyzed organized screening programs, via systematic doctor's prescriptions of tests targeting the at-risk population in order to ensure coverage, and those that assessed opportunistic screening through non-systematic test requests stemming from a doctor's appointment for other reasons or a routine checkup.

The primary outcome assessed was the case detection rate (CDR) of AMD in the early and intermediate stages. Early-stage disease was characterized by the presence of two medium-sized drusens (diameter >63 μm and ≤125 μm) with no pigmentary abnormalities and the intermediate stage by two large drusens (diameter > 125 μm) or pigmentary abnormalities [18,19]. The diagnostic accuracy of the screening tests, assessed based on sensitivity and specificity, was considered the secondary outcome.

## Information on sources and search strategy

The literature search was conducted on March 29, 2022, in the MEDLINE (via PubMed), Embase, Cochrane Library and Lilacs database. A manual search of the reference lists of eligible studies, Google Scholar and major journals on the topic was also carried out and, when necessary, the authors were contacted. The search terms used were "Macular Degeneration", "Maculopathy", "Macular Dystrophy", "Age-Related Macular Degeneration", "Wet Macular Degeneration", "Geographic Atrophy", "Dry Macular Degeneration", "Mass Screening", "Screening", "Vision Screening" and "opportunistic screening". The search strategies were adapted according to the specificities of each database (S1 Table in S1 File).

## Selection and data collection process

Study selection was performed independently by two pairs of researchers (BFL, DFG, DSPC and RMG). In the first stage, after removing duplicates, the researchers screened the first fifty studies and, based on the titles and abstracts, discussed any disagreements and adjusted the screening process. In stage two, the chosen studies were read in full and those that met the eligibility criteria were selected for data extraction and quality assessment. Disagreements in both stages were resolved by consensus or consulting a third researcher (ENS). Duplicates were removed by the administrator of Endnote references and selection was performed using Rayyan–Intelligent Systematic Review software.

Data extraction was carried out independently by the same four researchers (BFL, DFG, DSPC and RMG) using standardized spreadsheets with disagreements resolved by consensus. The data extracted included study characteristics such as screening setting, sample size, population characteristics, screening method, outcomes assessed, analysis characteristics and main findings.

## Risk of bias and quality of evidence assessment

Risk of bias and quality of evidence were evaluated independently by two researchers, with disagreements resolved by consensus. The ROBINS-I (Risk of Bias In Non-randomized Studies of Interventions) tool was used to assess non-randomized interventions [19]. Finally, GRADE (Grading of Recommendations Assessment, Development and Evaluation) was used to evaluate the quality of evidence [20,21].

## Data analysis and effect measures

The effect measures considered were the case detection rate (CDR) of the disease, mean differences with 95% confidence interval (CI), and relative risk with 95%CI, incidence or prevalence. The diagnostic accuracy of the screening tests, assessed based on sensitivity and specificity, was considered the secondary outcome. The significant differences in screening methods and outcomes assessed between studies precluded a meta-analysis and subgroup analysis of the studies included. As such, the results were presented descriptively.

## Results

### Study selection

The literature search resulted in 5,290 articles. After removing the duplicates, 4,550 were chosen to have their titles and abstracts analyzed. Fifty-two were considered potentially eligible and were read in full. Finally, three studies were included in the systematic review [22–24]. Fig 1 illustrates the selection process. The reasons for excluding articles are described in the supplementary material (S2 Table in S1 File).

### Study characteristics

Table 1 presents the main characteristics of the studies included in the systematic review. The three articles included [22–24] consist of three comparative observational studies. A total of 8,733 individuals (16,780 eyes) were included in the analysis. It is important to note that one study accounted for 92.3% of the individuals assessed [24], while the remaining two had relatively small samples [22,23]. The studies exhibited methodological heterogeneity, with varying population characteristics and screening settings. The average age of the study participants was older than 50 years.

The population of the three studies was submitted to organized screening programs, whereby they were invited to take part and undergo AMD screening tests [22–24]. The screening methods evaluated were based on OCT compared with color fundus photography (CFP) [24], OCT compared with a standard eye exam [22], and telemedicine testing compared with a standard eye exam [23].

### Clinical outcomes

Table 2 presents a summary of the studies selected, including a detailed description of the screening methods, outcomes assessed, screening characteristics, quality of evidence level and main outcomes. A retrospective observational study [24] assessed OCT compared with CFP and complementary use of both methods via population-based screening. The images collected were analyzed retrospectively to assess the diagnostic yield of the tests in identifying the clinical signs of AMD. Most of the OCT and CFP images were considered "gradable", that is, they ruled out the presence of AMD or classified the abnormalities identified by the tests (no abnormality, normal aging changes, early, intermediate or late AMD, or other retinal diseases). The results indicate that OCT provided gradable images in almost all the eyes examined (97.7%), whereas CFP provided a lower rate of gradable images in the eyes assessed (52.4%). Clinical signs of AMD were identified in 7.4% and 10.4% of the eyes and other retinal diseases in 10.8% and 8.7% for CFP and OCT, respectively [24]. In relation to the complementary use of both methods for AMD screening, the data suggest that OCT can be considered complementary to CFP, since the former performed better in classifying "gradable" images previously deemed "ungradable" or with no clinical signs by CFP [24] however; it is important to conduct studies whose design is suited to this evaluation. AMD was detected in 617/6,839 (7.4%) eyes assessed by CFP, with 3.9% graded as early AMD (grade 2), 2.3% intermediate (grade 3) and 1.2% late AMD (grade 4). In regard to the gradable OCT images, the disease was detected in 1,615 (10.4%) of the images, with 282 (1.8%) classified as late AMD (grade 2).

A prospective observational study [22] analyzed a set of screening tests for the detection of sight-threatening eye diseases, such as glaucoma, AMD and diabetic retinopathy in a cohort of elderly subjects recruited from primary care. In regard to AMD screening, OCT (full Retinal thickness), which enables examination of the macula and optic nerve, exhibited sensitivity and specificity of 50% and 73.4%, respectively. Sensitivity and specificity for visual acuity

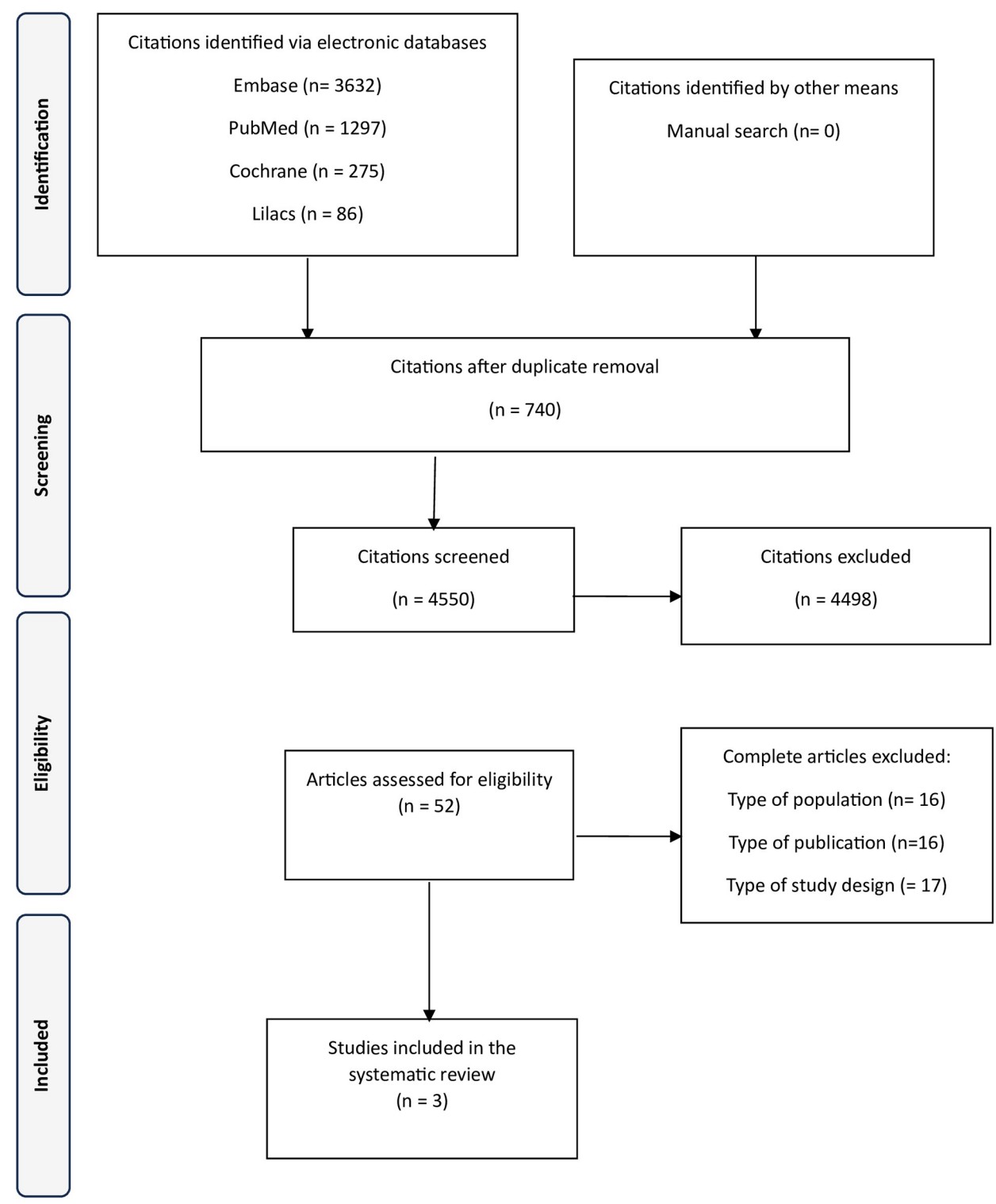

Figure 1. Flowchart of study selection.

**Fig 1. Flowchart of study selection.**

**Table 1. Characteristics of studies included in the systematic review.**

| Study | Type of study | Aim | Country | Setting | Sample size | Study conducted | Population characteristics | Type of screening | Quality assessment tool |
|---|---|---|---|---|---|---|---|---|---|
| Midena, 2020 [24] | Retrospective observational study | Analyze the contribution of OCT combined with CFP in AMD screening in a general unselected population (population-based screening) | Italy | Population-based screening | 8,069 individuals over 55 years old (15,957 eyes) | 2014–2015, images collected one a one-year period | The participants were voluntarily submitted to the screening tests | Screening program | ROBINS-I |
| Fidalgo, 2019 [22] | Prospective observational study | Determine the performance of a combination of screening tests in detecting eyesight-threatening diseases in a cohort of elderly subjects recruited in primary care | United Kingdom | Primary care | 505 individuals with an average age of 68 years | 2012–2013, the participants were invited to take part and included in the analysis over a one-year period | Individuals aged ≥60 years were recruited via a written invitation sent to community groups and a local optometry practice | Screening program | ROBINS-I |
| Hadziahmetovic, 2019 [23] | Prospective observational study | Test the feasibility and accuracy of a remote diagnostic imaging model using a combination of CFP and OCT (teleophthalmologya) as a clinical screening tool to facilitate identification of macular degeneration | United States | Primary care | 159 patients (318 eyes) | The remote imaging test and reference test were carried out 90 days apart | The study was conducted in locations with a high prevalence of diabetic retinopathy or AMD | Screening program | ROBINS-I |

Note: Age-related macular degeneration (AMD); Optical coherence tomography (OCT); color fundus photography (CFP); Risk Of Bias In Non-randomized Studies of Interventions (ROBINS-I) tool.

assessment (< 6/12) were 37.5% and 89.1%, and 60.4% and 64.3% for visual field analysis, respectively. Normality thresholds for the tests were prespecified and based on cutoff points. The data presented in the study were limited because of the lack of information about who conducted the reference examination. The results demonstrate that the tests can be useful in screening; however, multicentric studies with larger samples are needed to validate their use in screening for eye diseases [22].

One study investigated telemedicine-based screening for AMD via remote testing [23]. A prospective observational study evaluated a remote imaging screening model [23], whereby OCT and CFP results were analyzed remotely by trained but nonexpert imagers or in person by retinal specialists. The findings indicate that remote diagnostic imaging seemed to be equivalent to a standard examination in detecting AMD. Sensitivity was considered high (94%; 95% CI, 84–98%) for the individual or combined tests, where OCT exhibited greater specificity (93%; 95%CI, 87–96%) than CFP (63%; 95%CI, 53–71%). CFP obtained less favorable results due to the high rate of uninterpretable images (65.6% and 96.4% of the images analyzed by CFP and OCT, respectively, were interpretable), in addition to a lower positive predictive value when compared to OCT (56% vs 87%). The positive predictive value improved for combined OCT and CFP (89%). Both techniques exhibited high negative predictive values (CFP: 95%; OCT: 97% and combined CFP and OCT: 97%).

## Risk of bias and quality of evidence

Methodological quality was assessed using ROBINS-I, which classified one study as critical risk of bias due to confounding [24]; one as serious [23] and one as moderate risk of bias [22]. The risk of bias assessment of each study is summarized in Table 3.

**Table 2. Summary of the results of the included studies.**

| Author | Screening method (sample size) | Screening comparator method (sample size) | Outcomes | Analysis characteristics | Main findings | Quality assessment | Overall result |
|---|---|---|---|---|---|---|---|
| Midena, 2020 [24] | OCT (8,069 individuals/ 15,957 images) | CFP (8,069 individuals/ 15,957 images) | Rate of images classified as early and intermediate AMD | The images obtained by both screening techniques were analyzed anonymously by independent graders to prevent one analysis from influencing the other | **Rate of images (eyes) classified per AMD stageResults for gradable CFP images**• No clinical sign of AMD (grades 0 and 1): 81.8% (6,839 images) of the eyes.• Clinical signs of AMD identified in 7.4% (617 images): 3.9% classified as early AMD (grade 2); 2.3% intermediate (grade 3); and 1.2% late (grade 4). *Other retinal diseases were detected in 10.8% of the eyes examined (900 images, grade 5). **Results for gradable OCT images**• No clinical sign of AMD (grade 0): 80.9% (12,622 images) of the eyes.• Clinical signs of AMD identified in 10.4% (1,615 images) of the eyes: 1.8% late AMD (grade 2).*8.7% of the eyes exhibited other retinal diseases (grade 3).**Rate of identification of clinical signs of AMD based on the complementary use of screening methods** Signs of AMD were identified by OCT in 1,110 (6.9%) eyes whose CFP images were considered ungradable (847 eyes) or with no signs of AMD (263 eyes). On the other hand, CFP identified clinical signs of AMD in 157 (1.0%) eyes that showed no apparent signs of AMD in OCT. AMD was detected in 1,789 eyes (12.4%), considering gradable CFP and OCT images. | Critical | This study indicates that AMD screening should be based on the complementary use of at least two imaging exam formats. The results demonstrate that the use of OCT in AMD screening is complementary to CFP due to its high yield |

(*Continued*)

**Table 2.** (Continued)

| Author | Screening method (sample size) | Screening comparator method (sample size) | Outcomes | Analysis characteristics | Main findings | Quality assessment | Overall result |
|---|---|---|---|---|---|---|---|
| Fidalgo, 2019 [22] | OCT | Standard ophthalmic examination[a] (defined as visual acuity assessment, full anterior segment assessment by slit-lamp biomicroscopy, assessment of limbal anterior chamber depth by gonioscopy, posterior segment examination by indirect ophthalmoscopy and first-generation frequency doubling technology (FDT). | Sensitivity and specificity | An experienced ophthalmic technician performed all the screening tests with no prior knowledge of participants' ocular status or findings from the reference standard ophthalmic examination or OCT | **Sensitivity and specificity**<br>• Visual Acuity < 6/12 (Sensitivity: 37.5%; Specificity: 89.1%)<br>• FDT 1% Level (Sensitivity: 45.8%; Specificity: 77.7%)<br>• FDT 5% Level (Sensitivity: 60.4%; Specificity: 64.3%)<br>• SD-OCT (Full Retinal or GCC thickness) (Sensitivity: 52.1%; Specificity: 67.4%)<br>• SD-OCT (Full Retinal thickness) (Sensitivity: 50.0%; Specificity: 73.4%)<br>• SD-OCT (GCC thickness) (Sensitivity: 37.5%; Specificity: 85.9%)<br>• SD-OCT (Peripapillary RNFL thickness) (Sensitivity: 25.0%; Specificity: 93.0%) | Moderate | The set of screening tests provide more accurate and efficient population-based screening for significant eye diseases in older adults. This study provides useful preliminary data to inform the development of larger multicentric screening studies to validate this screening panel. |
| Hadziahmetovic, 2019 [23] | Remote diagnosis of imaging tests (telemedicine). The test was conducted using a device that incorporated CFP and OCT | Standard examination performed by a retinal specialist via OCT and CFP | Sensitivity and specificity Rate of images classified by disease stage | Remote diagnosis imaging was performed by trained but nonexpert imagers using a portable retinal imaging device that incorporated CFP and OCT. The standard examination was carried out by retinal specialists via OCT and CFP | **Remote diagnostic accuracy compared with the standard examination in identifying AMD**<br>OCT (sensitivity: 94%; specificity: 93%)<br>CFP (sensitivity: 94%; specificity: 63%)<br>Combined OCT and CFP (sensitivity: 94%; specificity: 95%)<br>**Disease classification:**<br>26 screened patients (10.4%) were identified as having nonproliferative diabetic retinopathy, with or without diabetic macular edema; 37 (14,8%) had intermediate or late AMD, including wet AMD. | Serious | Remote image assessment and standard examination by a retinal specialist were equivalent in identifying AMD in patients with a high prevalence of the disease. Additionally, combined OCT and CFP was associated with better operational outcomes |

Note: Optical coherence tomography (OCT); Color fundus photography (CFP); Age-related macular degeneration (AMD); Snellen chart best corrected visual acuity (BCVA).

[a] Participants in this study underwent a series of screening tests for various eye diseases. The following tests were assessed: (1) intraocular pressure (IOP); (2) first generation frequency doubling technology (FDT); and (3) structural examination of the macula and optical nerve by OCT. However, in this systematic review, only the results of the visual field test and structural examination of the macula and optical nerve were considered.

**Table 3. Methodological assessment of the studies included in this systematic review using the Risk of Bias In Non-randomized Studies of Interventions (ROBINS-I) tool.**

| Study | Bias due to confounding | Selection bias | Bias in classification of interventions | Bias due to deviations from intended interventions | Bias due to missing data | Bias in measurement of outcomes | Bias in selection of the reported result | Overall bias |
|---|---|---|---|---|---|---|---|---|
| **Midena, 2020** [24] | Critical | Moderate | Low | NI | Low | Moderate | Moderate | Critical |
| **Fidalgo, 2019** [22] | Moderate | Moderate | Low | Low | Low | Moderate | Moderate | Moderate |
| **Hadziahmetovic, 2019** [23] | Serious | Moderate | Low | Serious | Serious | Moderate | Moderate | Serious |

According to the GRADE method, evidence quality was considered very low for the outcome related to the AMD detection rate, evaluated by two studies, and for diagnostic accuracy (sensitivity and specificity), also assessed by two studies. For the outcomes analyzed, there were serious issues related to risk of bias and uncertainty (S3 Table in S1 File).

## Discussion

AMD can cause significant visual impairment and even blindness, although the early stages may be completely asymptomatic. Given its impact on the quality of life of patients, the associated economic burden and limited treatment options, it is important to discuss effective screening methods for detecting and classifying the disease. The results of this systematic review are limited and only suggest that AMD screening can be a useful tool to examine the retina and identify the disease in the at-risk population. Studies with adequate methodological quality will allow a satisfactory evaluation. The data on OCT suggest satisfactory performance in detecting the disease. The telemedicine-based tests showed promising results and were not inferior to conventional screening methods. However, the methodological problems and very low level of evidence mean that the data should be interpreted with caution. The data presented in this systematic review precluded inferring reliable conclusions about AMD screening. Furthermore, it is not possible to define any recommendation for decision makers.

One study included in this systematic review indicates that AMD screening should be based on the complementary use of at least two imaging exam formats and that the use of OCT in AMD screening is complementary to CFP due to its high yield [24]. This information suggests that multiple sequential tests can be used to screen for AMD. It is usually recommended that the first test be the least expensive and invasive and most tolerable, while the second should exhibit greater sensitivity and specificity than the first [25–27]. As such, there is a set of imaging tests that could help screen for and diagnose the disease.

Research indicates that tests such as OCT and OCT angiography (OCT-A) are useful and capable of detecting abnormalities not visible in CFP or fluorescein angiography [28]. Although OCT obtained the best results in this systematic review and can be considered the gold standard for AMD screening, it is still considered costly in relation to the alternatives [29]. Thus, it can be suggested that tests such as CFP may be recommended in the first stage of screening, but primary studies with better methodological design should be performed for this recommendation. According to Hadziahmetovic et al. [23], OCT is a more detailed examination recommended to confirm AMD diagnosis and exhibits greater specificity when compared to CFP.

One of the studies included in this systematic review [23] investigated telemedicine-based screening and reported it was not inferior to in-person screening by retinal specialists. However, the findings on AMD screening are incipient and indicate a need for further research with larger samples and better methodological quality to evaluate the reliability of screening

methods. An RCT registered with Clinical Trial is currently underway and could provide new insights for this debate, since it aims to evaluate a screening program compared with visual acuity testing alone in cases of AMD, glaucoma and diabetic retinopathy [30]. Furthermore, the use of artificial intelligence was not evaluated in this systematic review, but studies indicate that applying automated tools based on artificial intelligence can provide substantial benefits in the screening and diagnosis of AMD [31,32].

In the future, developing treatments to prevent AMD progression may improve the prognosis of the disease. Research is currently underway on preventive laser treatments for dry AMD, such as micropulse and nanosecond laser techniques [33–35]. The Early Stages of Age-Related Macular Degeneration (LEAD) study, for example, evaluated the safety of subthreshold nanosecond laser treatment in intermediate AMD and its efficacy for slowing progression to late AMD. The results found no significant reduction in the overall rate of progression to late AMD compared with sham treatment in intermediate AMD patients. However, post hoc analyses revealed a potential beneficial effect of subthreshold nanosecond laser (SNL) treatment in eyes without reticular pseudodrusen (RPD) at baseline, and that SNL treatment may increase the rate of progression to late AMD in eyes with RPD at baseline. As such, further studies are needed before recommendations can be made [36].

In terms of screening efficiency, an economic evaluation of chronic eye diseases indicated that defining and selecting the target population optimizes the cost-effectiveness of screening [37]. The authors of the study concluded that teleophthalmology was less costly for screening than the traditional clinical exam in cases of diabetic retinopathy and glaucoma. The main determining variable in the cost-effectiveness of teleophthalmology was the prevalence of retinopathy and glaucoma among the patients screened. Other factors that could potentially influence the cost-effectiveness of teleophthalmology were older patient age, regular screening and full use of the equipment [38].

Thus, new studies on the diagnostic accuracy of AMD screening tests and economic evaluations could provide support for health policies on the issue, similar to the case of diabetic retinopathy. Diabetic retinopathy screening programs have been implemented in several European countries in order to reduce the risk of visual impairment and blindness among asymptomatic individuals with diabetes by immediately identifying and effectively treating the condition [39]. However, in the case of diabetic retinopathy, early treatment is proven to be effective. In the United Kingdom, the National Health Service (NHS) invites people with diabetes aged 12 years and over to be screened once a year. They are referred to a hospital ophthalmology clinic for further testing and possible treatment if screening identifies eyesight-threatening signs of diabetic retinopathy [40,41]. In this case, screening is an important tool to mitigate the social and economic impacts of failing to provide timely patient care and reduce the possibility of irreversible blindness. In the case of AMD, studies suggest that vitamin and mineral supplements can delay progression of the disease to the advanced stage, but scientific evidence about efficacy and safety is limited [12,13]. As such, the NICE guideline does not recommend their use for AMD [14]. Therefore, this systematic review aimed to assess whether AMD screening could be considered effective in identifying patients with early clinical signs of the disease should treatments for these AMD stages available in the future, as well as those who have progressed to advanced AMD for early treatment prior to their self-presentation.

The heterogeneity of the methods and outcomes assessed and the lack of information on establishing a nomenclature to classify the disease precluded comparing the results of the studies included in this systematic review. According to expert consensus, each imaging method is a source of data that can provide independent information on the structure or physiology of the macula and enable simultaneous understanding of the physiological characteristics of the disease [28]. Although the use of multiple imaging methods contributed to different points of

view on the molecular pathological characteristics of the disease, this expanded analytical capacity could produce a fragmented conceptualization of the disease and varied classification [28,42]. As such, a standardized nomenclature allows the results of research to be compared using imaging methods by evaluating multimodal imaging concepts and known histopathological characteristics.

The main limitation of this review is the low to moderate methodological quality of the studies included and the very low evidence quality for the outcomes of interest (case detection rate and diagnostic accuracy). Only one study provided data on thresholds of normality for the screening test results [22] and two presented data on the classification of early and intermediate-stage disease following screening [23,24]. No data were provided on the periodicity of screening tests. Another limitation was the heterogeneity of the studies, which limited comparison of the results. Additionally, new studies about this topic should assess the potential harm to patients of AMD screening, including overdiagnosis and psychosocial effects such as anxiety, sadness and sleep problems [10]. However, it should be noted that the literature on the topic is incipient and still scarce. Although the findings are preliminary, it is important to underscore that the data presented in this systematic review can provide support for future discussion on the issue.

## Conclusion

The data presented in this systematic review precluded inferring reliable conclusions about AMD screening. Thus, the systematized data are limited and only suggest satisfactory performance in early screening of the population at risk of developing AMD. Although the results presented seem promising, methodological problems were identified in the studies selected and the level of evidence was considered very low. As such, the results should be interpreted with caution. OCT and the telemedicine technique showed promising results in AMD screening. However, studies with larger samples and better methodological quality are needed to assess the performance of screening tools and evaluate and compare the multiple tests that might be applied sequentially to ensure adequate screening. Additionally, future research on the cost and periodicity of screening could prompt more robust debate on the possible implementation of AMD screening programs.

It should be noted that the need to discuss strategies that equitably maximize the health and opportunities of people with visual impairments as they age is a challenge, particularly in low and middle-income countries which still need to resolve longstanding issues concomitantly to new challenges. Thus, public policies are needed to meet the healthcare demand, which is driven by population aging and demographic transition. Screening for eye diseases that can cause irreversible blindness, especially AMD, is a tool that could mitigate the social and economic impacts of the disease.

## Supporting information

**S1 Checklist. PRISMA 2020 checklist.**
(DOCX)

**S1 File. Supplementary material–contains supporting tables.**
(DOCX)

## Author Contributions

**Conceptualization:** Dalila Fernandes Gomes, Daniel da Silva Pereira Curado, Rosângela Maria Gomes, Betânia Ferreira Leite, Everton Nunes da Silva.

**Formal analysis:** Dalila Fernandes Gomes, Daniel da Silva Pereira Curado, Rosângela Maria Gomes, Betânia Ferreira Leite, Everton Nunes da Silva.

**Methodology:** Dalila Fernandes Gomes, Daniel da Silva Pereira Curado, Rosângela Maria Gomes, Betânia Ferreira Leite, Everton Nunes da Silva.

**Supervision:** Dalila Fernandes Gomes, Everton Nunes da Silva.

**Validation:** Dalila Fernandes Gomes, Daniel da Silva Pereira Curado, Rosângela Maria Gomes, Betânia Ferreira Leite, Everton Nunes da Silva.

**Writing – original draft:** Dalila Fernandes Gomes.

**Writing – review & editing:** Dalila Fernandes Gomes, Daniel da Silva Pereira Curado, Rosângela Maria Gomes, Betânia Ferreira Leite, Maíra Catharina Ramos, Everton Nunes da Silva.

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
