## [Decision Letter · Decision Letter 0]

2 Aug 2023

PONE-D-23-03440Clinical effectiveness of screening for age-related macular degeneration: a systematic reviewPLOS ONE

Dear Dr. Gomes,

Thank you for submitting your manuscript to PLOS ONE. After careful consideration, we feel that it has merit but does not fully meet PLOS ONE’s publication criteria as it currently stands. Therefore, we invite you to submit a revised version of the manuscript that addresses the points raised during the review process.

We look forward to receiving your revised manuscript.

Kind regards,

Andrzej Grzybowski

Academic Editor

PLOS ONE

Reviewers' comments:

Reviewer's Responses to Questions

**Comments to the Author**

1. Is the manuscript technically sound, and do the data support the conclusions?

Reviewer #1: No

Reviewer #2: Yes

2. Has the statistical analysis been performed appropriately and rigorously? 

Reviewer #1: N/A

Reviewer #2: Yes

3. Have the authors made all data underlying the findings in their manuscript fully available?

Reviewer #1: Yes

Reviewer #2: Yes

4. Is the manuscript presented in an intelligible fashion and written in standard English?

Reviewer #1: Yes

Reviewer #2: Yes

5. Review Comments to the Author

Reviewer #1: 1. I am interested in why the authors were keen to review the literature on screening for early AMD. Early AMD is asymptomatic and has no treatment nor is likely to in the near future, and so there is no benefit from identifying it. The more useful question to clinicians is can we screen people flor late AMD which has treatment or treatment is likely in the near future.

2. Relatedly, the paper has no introduction or discussion of the principles of screening.

3. The conclusions are not supported by the papers and data included in the review eg Line 249. The review includes weak, sparce and biased data, so the conclusion that screening is useful is unwarranted.

Other points.

4. By definition AMD affects people over and age threshold, usually taken as >55yrd but sometimes >50yrs. People under that age group have something else such as idiopathic macular degeneration or macular dystrophy- lines 30 and 60.

5. In high-income countries AMD IS THE leading cause of blindness. Line 65.

6. Line 75. It is usually when the non-dominant eye is affected patients’ do not notice visual loss. It would seem to be the fact that non-dominance exists rather than a compensatory mechanism by the brain.

7. Line 71. A reference for this data please.

8. Why was CDR chosen as the primary outcome measure? This is useful if the prevalence in the target population is known. It gives no information on the effectiveness of a screening intervention.

9. Line 154. Can the authors justify use of odds ratio here. Not a standard assessment for screening interventions.

10. Study ref 19 cannot be considered as screening. It is an evaluation of triaging referrals to a retinal clinic and assessment of referral accuracy. The population are referred patients so this study does not meet the reviews inclusion criteria of studies investigating screening.

11. Line 194. The study does not show that OCT is mor accurate than CFP because there was no assessment against a gold standard and no assessment of false positives.

12. Line 197. The study does not indicate that the methods are complimentary, that is an inference from the data which suggests that might be the case but does not demonstrate it.

13. Line209. Ref 17. It is not clear who conducted the reference examination and if this was taken as the gold standard. In fact the sens and spec of the individual tests are much too low to be used as screening tests. Line 210-11 ‘more accurate and efficient ‘than what? There does not seem to be any assessment of efficiency – cost/episode- in this study. Line 214 The results do not demonstrate that the tests are useful in screening because the S&S were too low.

14. Line 217. Did the retinal specialist do funduscopy or just look at images?

15. Line248. Convenience was not reported to be assessed by any of the studies so no conclusions about convenience can be made.

16. Line 249. In order to be useful tests need to be accurate. The was either no assessment of accuracy or it was found to be ltoo low for screening tests.

17. There is no discussion of harms that can arise from screening such as increasing anxiety in a population with a low overall risk of vision loss.

18. Line 252. I would say the methological problems mean we cannot draw any conclusions from the studies available. Of the 3 appropriate studies 2 had critical or serious risk of bias and one had moderate risk but found low S&S.

19. Line 256. Screening inevitably increases the workload of clinicians, at least initially, as more cases are found.

20. Lone 261 There is no datat here to suggest retesting would reduce false positives.

21. Line 288. DR is a different situation where early treatment is proven to be effective unlike early AMD.

22. Line323. ‘must' is too strong here. The data is suggestive but unproven.

Reviewer #2: This is a comprehensive study concerning the effectiveness of early screening in recognizing AMD. The manuscript itself is properly written and organized, with clear tables and figures.

Study is limited by the evidency quality of the assessed studies, as authors state in the limitations part.

My only concern is why the artifical intelligence was excluded completely from the study. As this is an emerging trend now it would be beneficial if authors would elaborate on that in the exclusion criteria, and add a certai paragraph in the discussion part about pros and cons of this approach.

6. PLOS authors have the option to publish the peer review history of their article (what does this mean?). If published, this will include your full peer review and any attached files.

Reviewer #1: **Yes: **Nicholas Beare

Reviewer #2: No

---

## [Author Response · Author response to Decision Letter 0]

15 Sep 2023

Reviewer #1, comment 1: “1. I am interested in why the authors were keen to review the literature on screening for early AMD. Early AMD is asymptomatic and has no treatment nor is likely to in the near future, and so there is no benefit from identifying it. The more useful question to clinicians is can we screen people flor late AMD which has treatment or treatment is likely in the near future.”

Our response: We would like to thank you for your consideration and time spent reviewing our manuscript. Your comments allow us to improve the manuscript. 

The early stage of AMD is asymptomatic and disease progression can occur rapidly. In some cases, patients do not notice the development of visual impairment until the disease has progressed to an advanced stage. Therefore, this systematic review aimed to assess whether screening individuals with early-stage AMD could be useful for clinical patient monitoring and, if necessary, carrying out appropriate timely treatment. In the introduction was included a new paragraph with this justification, as described below: 

“According to the World Health Organization (WHO), the purpose of screening is to identify people in an apparently healthy population who are at higher risk of a health problem or a condition, so that an early treatment or intervention can be offered [10, 11]. In some cases, such as antenatal screening, the aim is to provide information about an increased risk or condition to help people make an informed decision about their care or treatment[10]. Currently, there are no medical treatments for early or intermediate AMD, only evidence that antioxidant vitamin and mineral supplementation can delay progression to the advanced stage and the loss of visual acuity in people with signs of the disease [12, 13]. In this scenario, the present systematic review evaluated whether AMD screening could be considered a tool for clinically monitoring patients with early clinical signs of the disease and, if necessary, starting timely treatment for those in an advanced stage”.

Reviewer #1, comment 2: “2. Relatedly, the paper has no introduction or discussion of the principles of screening.”

Our response: The manuscript was updated. In the introduction was included a new paragraph about the principles of screening and the reason for preparing this systematic review.

“According to the World Health Organization (WHO), the purpose of screening is to identify people in an apparently healthy population who are at higher risk of a health problem or a condition, so that an early treatment or intervention can be offered [10, 11]. In some cases, such as antenatal screening, the aim is to provide information about an increased risk or condition to help people make an informed decision about their care or treatment[10]. Currently, there are no medical treatments for early or intermediate AMD, only evidence that antioxidant vitamin and mineral supplementation can delay progression to the advanced stage and the loss of visual acuity in people with signs of the disease [12, 13]. In this scenario, the present systematic review evaluated whether AMD screening could be considered a tool for clinically monitoring patients with early clinical signs of the disease and, if necessary, starting timely treatment for those in an advanced stage”.

Reviewer #1, comment 3: “3. The conclusions are not supported by the papers and data included in the review eg Line 249. The review includes weak, sparce and biased data, so the conclusion that screening is useful is unwarranted.”

Our response: We agree that the studies included in this systematic review have serious methodological limitations. Therefore, the manuscript was updated and the information below was described in the text:

“The results of this systematic review are limited and only suggest that AMD screening can be a useful tool to examine the retina and identify the disease in the at-risk population. Studies with adequate methodological quality will allow a satisfactory evaluation. The data on OCT suggest satisfactory performance in detecting the disease. The telemedicine-based tests showed promising results and were not inferior to conventional screening methods. However, the methodological problems and very low level of evidence mean that the data should be interpreted with caution. The data presented in this systematic review precluded inferring reliable conclusions about AMD screening. Furthermore, it is not possible to define any recommendation for decision makers”.

Reviewer #1, comment 4: “Other points. 4. By definition AMD affects people over and age threshold, usually taken as >55yrd but sometimes >50yrs. People under that age group have something else such as idiopathic macular degeneration or macular dystrophy- lines 30 and 60.”

Our response: This systematic review defined among the eligibility criteria the inclusion of studies that evaluated people from 50 years of age for the extensive screening of scientific evidence. The definition of this age group is supported by bibliographic reference presented in the text.

Reference: Jager RD, Mieler WF, Miller JW (2008) Age-related macular degeneration. New England Journal of Medicine 358:2606–2617

Reviewer #1, comment 5: “5. In high-income countries AMD IS THE leading cause of blindness. Line 65.”

Our response: Thanks for the comment. The manuscript has been corrected as suggested.

Reviewer #1, comment 6: “6. Line 75. It is usually when the non-dominant eye is affected patients’ do not notice visual loss. It would seem to be the fact that non-dominance exists rather than a compensatory mechanism by the brain.”

Our response: Thanks for the comment. Some studies describe that because of the brain compensation mechanisms, the patient may not notice any change in vision in the early stages of the disease. This information is supported by the following studies:

• SCHWARTZ, Roy; LOEWENSTEIN, Anat. Early detection of age related macular degeneration: current status. International Journal of Retina and Vitreous, v. 1, n. 1, p. 1-8, 2015.

• BRESSLER, Neil M. Early detection and treatment of neovascular age-related macular degeneration. The Journal of the American Board of Family Practice, v. 15, n. 2, p. 142-152, 2002.

The reviewer's comment allows us to understand that other reasons may delay diagnosis in some cases, such as when the non-dominant eye is affected and therefore patients’ do not notice visual loss. Therefore, we included in the manuscript the description that the involvement of non-dominant eye can also delay the diagnosis of AMD and we include a reference to this information, as described below: 

“Early diagnosis enables adequate treatment and a better prognosis, but can be delayed for different reasons, such as the fact that the disease is asymptomatic in its early stages; compensatory mechanisms in the brain that make it difficult for patients to notice changes in vision in the initial stages; involvement of non-dominant eye and lack of awareness about the disease[7–9]”.

• Reference about non-dominant eye: Rai BB, Shresthra MK, Thapa R, Essex RW, Paudyal G, Maddess T. Pattern and presentation of vitreo-retinal diseases: an analysis of retrospective data at a tertiary eye care center in Nepal. Asia-Pac J Ophthalmol. 2019;8(6):481–488.

Reviewer #1, comment 7: “7. Line 71. A reference for this data please.”

Our response: The reference for this data was included in the manuscript. The description about the expected increase of cases between 2020 and 2040 was based on the study below and the description of the global prevalence of AMD in people aged 45 to 85 years was based on the Supplementary Material of the same study. Therefore, we included in the manuscript the main reference of the study and the reference of the supplementary material.

References:

• WONG, Wan Ling et al. Global prevalence of age-related macular degeneration and disease burden projection for 2020 and 2040: a systematic review and meta-analysis. The Lancet Global Health, v. 2, n. 2, p. e106-e116, 2014.

• WONG, Wan Ling et al. Supplementary appendix. Global prevalence of age-related macular degeneration and disease burden projection for 2020 and 2040: a systematic review and meta-analysis. The Lancet Global Health, v. 2, n. 2, p. e106-e116, 2014.

Reviewer #1, comment 8: “8. Why was CDR chosen as the primary outcome measure? This is useful if the prevalence in the target population is known. It gives no information on the effectiveness of a screening intervention.”

Our response: The case detection rate (CDR) of AMD was included as primary outcome because in the study planning stage the authors of this systematic review deduced that this would be the main outcome reported in the studies included in the systematic review. On this basis, we kept this outcome to comply with the research protocol of our systematic review registered in Prospero repository. Additionally, other systematic reviews have used CDR as primary outcome, such as the reference below:

• FARIA, Lídia et al. Gastric cancer screening: a systematic review and meta-analysis. Scandinavian Journal of Gastroenterology, v. 57, n. 10, p. 1178-1188, 2022.

Reviewer #1, comment 9: “9. Line 154. Can the authors justify use of odds ratio here. Not a standard assessment for screening interventions.”

Our response: Thank you for your comment. We removed “odds ratio” from the text as this measure is not appropriate to screening intervention. 

Reviewer #1, comment 10: “10. Study ref 19 cannot be considered as screening. It is an evaluation of triaging referrals to a retinal clinic and assessment of referral accuracy. The population are referred patients so this study does not meet the reviews inclusion criteria of studies investigating screening.”

Our response: Thank for raising this point. In fact, the inclusion of this study was largely discussed by the authors of this systematic review. We have raised pros and cons about this inclusion, opting for keeping it in the systematic review. However, as you have pointed out, it is clearer for us now that it does not comply with our eligibility criteria. Based on that, we excluded this study from our systematic review. 

Reviewer #1, comment 11: “11. Line 194. The study does not show that OCT is mor accurate than CFP because there was no assessment against a gold standard and no assessment of false positives.”

Our response: We agree with the reviewer. The manuscript was updated with the following information: “The results indicate that OCT provided gradable images in almost all the eyes examined (97.7%), whereas CFP provided a lower rate of gradable images in the eyes assessed (52.4%)”.

Reviewer #1, comment 12: “12. Line 197. The study does not indicate that the methods are complimentary, that is an inference from the data which suggests that might be the case but does not demonstrate it.”

Our response: We agree that the study has limitations related to this issue. We updated the text with the following information: 

“In relation to the complementary use of both methods for AMD screening, the data suggest that OCT can be considered complementary to CFP, since the former performed better in classifying “gradable” images previously deemed “ungradable” or with no clinical signs by CFP[24] however; it is important to conduct studies whose design is suited to this evaluation”.

Reviewer #1, comment 13: “13. Line209. Ref 17. It is not clear who conducted the reference examination and if this was taken as the gold standard. In fact the sens and spec of the individual tests are much too low to be used as screening tests. Line 210-11 ‘more accurate and efficient ‘than what? There does not seem to be any assessment of efficiency – cost/episode- in this study. Line 214 The results do not demonstrate that the tests are useful in screening because the S&S were too low.”

Our response: We agree with the reviewer, the lack of information about who conducted the reference examination and whether this was taken as the gold standard limited the evaluation of the screening method described in the study. This information was described in the text: “The data presented in the study were limited because of the lack of information about who conducted the reference examination”.

The information about ‘more accurate and efficient’ was deleted from the text.

Other information was updated in this paragraph: “The results demonstrate that the tests can be useful in screening; however, multicentric studies with larger samples are needed to validate their use in screening for eye diseases [22]”.

Reviewer #1, comment 14: 14. Line 217. Did the retinal specialist do funduscopy or just look at images?”

Our response: Thank you for raising this point. We reassessed the study and this information is not clear. The only information available is that retina specialist performed the standard ophthalmologic examination.

Reviewer #1, comment 15: “15. Line248. Convenience was not reported to be assessed by any of the studies so no conclusions about convenience can be made.”

Our response: The studies included in the systematic review have many methodological limitations as described in the manuscript and limit the conclusions. The text was updated: “The results of this systematic review are limited and only suggest that AMD screening can be a useful tool to examine the retina and identify the disease in the at-risk population”.

Reviewer #1, comment 16: “16. Line 249. In order to be useful tests need to be accurate. The was either no assessment of accuracy or it was found to be ltoo low for screening tests.”

Our response: We agree with the reviewer. The manuscript was updated with the following information: 

“The results of this systematic review are limited and only suggest that AMD screening can be a useful tool to examine the retina and identify the disease in the at-risk population. Studies with adequate methodological quality will allow a satisfactory evaluation. The data on OCT suggest satisfactory performance in detecting the disease. The telemedicine-based tests showed promising results and were not inferior to conventional screening methods. However, the methodological problems and very low level of evidence mean that the data should be interpreted with caution. The data presented in this systematic review precluded inferring reliable conclusions about AMD screening. Furthermore, it is not possible to define any recommendation for decision makers”.

Reviewer #1, comment 17: “17. There is no discussion of harms that can arise from screening such as increasing anxiety in a population with a low overall risk of vision loss.”

Our response: Thanks for the comment. The manuscript was updated with information about the potential harms of screening, as described below:

“Additionally, new studies about this topic should assess the potential harm to patients of AMD screening, including overdiagnosis and psychosocial effects such as anxiety, sadness and sleep problems[10]”.

Reviewer #1, comment 18: “18. Line 252. I would say the methological problems mean we cannot draw any conclusions from the studies available. Of the 3 appropriate studies 2 had critical or serious risk of bias and one had moderate risk but found low S&S.”

Our response: We thank you for your comment and agree that the data presented in the review are only suggestive, not confirmatory. All limitations identified in this systematic review are described in the manuscript, including methodological limitations and low-quality evidence. Additionally, we describe in the text that the results should be interpreted with caution.

The manuscript has been updated with the following information: “The data presented in this systematic review precluded inferring reliable conclusions about AMD screening. Furthermore, it is not possible to define any recommendation for decision makers”.

Reviewer #1, comment 19: “19. Line 256. Screening inevitably increases the workload of clinicians, at least initially, as more cases are found.”

Our response: As there is no evidence on convenience from the included studies, we removed this information in the discussion.

Reviewer #1, comment 20: “20. Lone 261 There is no datat here to suggest retesting would reduce false positives.”

Our response: This information was deleted and the text updated:

“One study included in this systematic review indicates that AMD screening should be based on the complementary use of at least two imaging exam formats and that the use of OCT in AMD screening is complementary to CFP due to its high yield[24]. This information suggests that multiple sequential tests can be used to screen for AMD. It is usually recommended that the first test be the least expensive and invasive and most tolerable, while the second should exhibit greater sensitivity and specificity than the first[26–28]. As such, there is a set of imaging tests that could help screen for and diagnose the disease”.

Reviewer #1, comment 21: “21. Line 288. DR is a different situation where early treatment is proven to be effective unlike early AMD.”

Our response: Thanks for the comment. The manuscript was updated and described that in the case of diabetic retinopathy, early treatment is proven to be effective. In the case of AMD, only the use of vitamin supplements can be recommended for early treatment, but this systematic review aimed to assess whether screening individuals with early-stage AMD could be useful for clinical patient monitoring and, if necessary, carrying out appropriate timely treatment. The text was updated as described below:

“Thus, new studies on the diagnostic accuracy of AMD screening tests and economic evaluations could provide support for health policies on the issue, similar to the case of diabetic retinopathy. Diabetic retinopathy screening programs have been implemented in several European countries in order to reduce the risk of visual impairment and blindness among asymptomatic individuals with diabetes by immediately identifying and effectively treating the condition[36]. However, in the case of diabetic retinopathy, early treatment is proven to be effective. In the United Kingdom, the National Health Service (NHS) invites people with diabetes aged 12 years and over to be screened every 3, 6, 9 or 12 months, depending on specific eye changes and how fast they occur. They are referred to a hospital ophthalmology clinic for further testing and possible treatment if screening identifies eyesight-threatening signs of diabetic retinopathy[37]. In this case, screening is an important tool to mitigate the social and economic impacts of failing to provide timely patient care and reduce the possibility of irreversible blindness. In the case of AMD, only vitamin and mineral supplements can be recommended in early treatment to delay progression of the disease to the advanced stage[12, 13]. Therefore, this systematic review aimed to assess whether screening individuals with early-stage AMD could be useful for clinical patient monitoring and, if necessary, carrying out appropriate timely treatment”.

Reviewer #1, comment 22: “22. Line323. ‘must' is too strong here. The data is suggestive but unproven.”

Our response: Thanks for the comment. The term was changed in the manuscript to 'might'.

Reviewer #2, comment 1: “This is a comprehensive study concerning the effectiveness of early screening in recognizing AMD. The manuscript itself is properly written and organized, with clear tables and figures. Study is limited by the evidency quality of the assessed studies, as authors state in the limitations part.”

Our response: We would like to thank you for your consideration and time spent reviewing our manuscript. Your comments allow us to improve the manuscript. 

Reviewer #2, comment 2: “My only concern is why the artifical intelligence was excluded completely from the study. As this is an emerging trend now it would be beneficial if authors would elaborate on that in the exclusion criteria, and add a certai paragraph in the discussion part about pros and cons of this approach.”

Our response: The use of artificial intelligence was excluded from this systematic review because in the planning stage of this study a systematic review on this topic was identified (published in 2021). Therefore, this type of evaluation was not included in this systematic review.

We included in the discussion information about studies that evaluated the use of artificial intelligence in screening and diagnosis of AMD, as described below:

“Furthermore, the use of artificial intelligence was not evaluated in this systematic review, but studies indicate that applying automated tools based on artificial intelligence can provide substantial benefits in the screening and diagnosis of AMD[32, 33]”.

---

## [Decision Letter · Decision Letter 1]

5 Oct 2023

PONE-D-23-03440R1Clinical effectiveness of screening for age-related macular degeneration: a systematic reviewPLOS ONE

Dear Dr. Gomes,

Thank you for submitting your manuscript to PLOS ONE. After careful consideration, we feel that it has merit but does not fully meet PLOS ONE’s publication criteria as it currently stands. Therefore, we invite you to submit a revised version of the manuscript that addresses the points raised during the review process.

We look forward to receiving your revised manuscript.

Kind regards,

Andrzej Grzybowski

Academic Editor

PLOS ONE

Journal Requirements:

Reviewers' comments:

Reviewer's Responses to Questions

**Comments to the Author**

1. If the authors have adequately addressed your comments raised in a previous round of review and you feel that this manuscript is now acceptable for publication, you may indicate that here to bypass the “Comments to the Author” section, enter your conflict of interest statement in the “Confidential to Editor” section, and submit your "Accept" recommendation.

Reviewer #1: (No Response)

Reviewer #3: All comments have been addressed

2. Is the manuscript technically sound, and do the data support the conclusions?

Reviewer #1: Yes

Reviewer #3: Yes

3. Has the statistical analysis been performed appropriately and rigorously? 

Reviewer #1: N/A

Reviewer #3: Yes

4. Have the authors made all data underlying the findings in their manuscript fully available?

Reviewer #1: Yes

Reviewer #3: Yes

5. Is the manuscript presented in an intelligible fashion and written in standard English?

Reviewer #1: Yes

Reviewer #3: Yes

6. Review Comments to the Author

Reviewer #1: Overall I think the authors have addressed my concerns and improved the review substantially. My main remaining concern is by the addition of information about AREDS supplementation they have failed to present the controversy regarding whether AREDS supplementation is of any benefit in AMD (see below)

Comment 1.

I appreciate your added explanation with regards to screening. However the AREDS trial did not show any benefit for people with early AMD which is why they did a sub-group analysis on people with Intermediate AMD at high risk of progression. Given the positive finding was a post-hoc sub-group analysis, whether AREDS supplements are of any benefit in AMD is controversial. And although AREDS purported to show a reduced risk of progression, there was no significant difference in vision between the groups. Many consider the evidence of benefit too weak to give a firm clinical recommendation on AREDS which is why AREDS supplements are not recommended in the UK by national guidelines (NICE AMD Guidelines 2018). I suggest editing this added paragraph to reflect this uncertainty around AREDS.

In the final sentence the authors mix screening and clinical monitoring which are different objectives. I suggest that the systematic review evaluated whether AMD screening could be considered effective for identifying patients with early clinical signs of the disease in case treatments for these stages of AMD become available in the future, and identifying patients with advanced AMD for early treatment prior to their self-presentation.

Comment 3. Excellent re-write.

Comment 4.

My point here is that line 38 “(AMD) is an eye disease most prevalent in patients over 50 years old” should read “…is an eye disease that occurs in patients over 50 years old”, and line 60 “that generally affects patients over the age of 50” should read “that affects patients over the age of 50”. Because is only affects patients over 50 years.

Comment 21.

I am generally happy with this re-write but in the UK the NHS invites people with diabetes to screening only every 12 months. If more frequent monitoring is required they are referred to ophthalmology clinics.

And see comments in Comment 1 above about AREDS.

Reviewer #3: • Evaluation of revision : authors responded carefully to all comments of both reviewers’ especially reviewer 1. The manuscript was very much improved after corrections and amendments made by the authors.

• Despite limited material available , the study provides important information about , in fact, lack of reliable and consistent screening approach to early and intermediate AMD. In the context of new technologies, as mentioned by reviewer 2, the burden of workload for AMD screening might be significantly reduced in the future and implementation of screening protocols based on AI made easily accessible.

• My minor remark:

a. Please mention preventive laser treatments for dry AMD (micropulse laser, 2-RT laser). Their efficacy is questioned at the moment, but the research is going on. If they have proven really preventive, than early screening would be justified.

7. PLOS authors have the option to publish the peer review history of their article (what does this mean?). If published, this will include your full peer review and any attached files.

Reviewer #1: **Yes: **Nicholas Beare

Reviewer #3: No

---

## [Author Response · Author response to Decision Letter 1]

30 Oct 2023

Reviewer #1: “Overall I think the authors have addressed my concerns and improved the review substantially. My main remaining concern is by the addition of information about AREDS supplementation they have failed to present the controversy regarding whether AREDS supplementation is of any benefit in AMD (see below).”

Reviewer #1, comment 1: “I appreciate your added explanation with regards to screening. However the AREDS trial did not show any benefit for people with early AMD which is why they did a sub-group analysis on people with Intermediate AMD at high risk of progression. Given the positive finding was a post-hoc sub-group analysis, whether AREDS supplements are of any benefit in AMD is controversial. And although AREDS purported to show a reduced risk of progression, there was no significant difference in vision between the groups. Many consider the evidence of benefit too weak to give a firm clinical recommendation on AREDS which is why AREDS supplements are not recommended in the UK by national guidelines (NICE AMD Guidelines 2018). I suggest editing this added paragraph to reflect this uncertainty around AREDS.”

Our response: As suggested, the paragraph was edited and described that scientific evidence on the efficacy and safety of use antioxidant vitamin and mineral supplementation is limited based on information from the AREDS study, as described below:

“Currently, there are no medical treatments for early or intermediate AMD, only evidence that antioxidant vitamin and mineral supplementation can delay progression to the advanced stage and the loss of visual acuity in people with signs of the disease. However, the scientific evidence is limited and there are several unanswered questions regarding antioxidant vitamin and mineral supplementation in the prevention of AMD, including at what stage the protective effect may be important and potential interactions with genetic effects and other risk factors for the disease, such as smoking. Furthermore, the safety of antioxidant vitamin and mineral supplementation is controversial[12, 13]”. 

“In 2018, the National Institute for Health and Care Excellence (NICE) guideline does not recommend using antioxidant and zinc supplements for AMD. According to this guideline, although the AREDS study showed some beneficial effects of combined antioxidant supplementation in a subgroup of participants, the effects of the individual formula components on AMD progression were unclear and one of the ingredients (beta carotene) was associated with a possible risk of lung cancer among smokers. The guideline also highlights that although the AREDS research group introduced a new formulation that excluded beta carotene in the AREDS2 study, study design limitations related to secondary randomization and no placebo control mean that the effect of this formulation on AMD disease progression remains unknown. A well-conducted randomized trial could therefore provide new evidence on the benefits and risks of individual components of antioxidant supplements [14]”.

Reviewer #1, comment 2: “In the final sentence the authors mix screening and clinical monitoring which are different objectives. I suggest that the systematic review evaluated whether AMD screening could be considered effective for identifying patients with early clinical signs of the disease in case treatments for these stages of AMD become available in the future, and identifying patients with advanced AMD for early treatment prior to their self-presentation”.

Our response: We appreciate the reviewer's comment. The objective of the study was updated, as described below: 

“In this scenario, the present systematic review evaluated whether AMD screening could be considered effective in identifying patients with early clinical signs of the disease should treatments for these stages of AMD become available in the future, as well as those who have progressed to advanced AMD for early treatment prior to their self-presentation.”

Reviewer #1, comment 3. Excellent re-write.

Our response: Thank you for the comment.

Reviewer #1, comment 4. “My point here is that line 38 “(AMD) is an eye disease most prevalent in patients over 50 years old” should read “…is an eye disease that occurs in patients over 50 years old”, and line 60 “that generally affects patients over the age of 50” should read “that affects patients over the age of 50”. Because is only affects patients over 50 years.”

Our response: Thank you for the comment. The manuscript was updated with the following information: 

• “Age-related macular degeneration (AMD) is an eye disease that occurs in patients over 50 years old.”

• “Age-related macular degeneration (AMD) is a chronic progressive disease that affects patients over the age of 50”.

Reviewer #1, comment 5 (related to comment 21 from the previous revision). “I am generally happy with this re-write but in the UK the NHS invites people with diabetes to screening only every 12 months. If more frequent monitoring is required they are referred to ophthalmology clinics. And see comments in Comment 1 above about AREDS”.

Our response: Thank you for the comment about screening for diabetic retinopathy in the NHS. The information about screening every 3, 6, 9 or 12 months was updated in the text and included a new reference (NICE Guideline 2018). The manuscript was updated with the following information: “In the United Kingdom, the National Health Service (NHS) invites people with diabetes aged 12 years and over to be screened once a year”.

In the same paragraph, the manuscript was also updated with the following information: “In the case of AMD, studies suggest that vitamin and mineral supplements can delay progression of the disease to the advanced stage, but scientific evidence about efficacy and safety is limited [12, 13]. As such, the NICE guideline does not recommend their use for AMD[14]. Therefore, this systematic review aimed to assess whether AMD screening could be considered effective in identifying patients with early clinical signs of the disease should treatments for these AMD stages available in the future, as well as those who have progressed to advanced AMD for early treatment prior to their self-presentation”.

Reviewer #3, comment 1: “Evaluation of revision: authors responded carefully to all comments of both reviewers’ especially reviewer 1. The manuscript was very much improved after corrections and amendments made by the authors”.

• “Despite limited material available, the study provides important information about, in fact, lack of reliable and consistent screening approach to early and intermediate AMD. In the context of new technologies, as mentioned by reviewer 2, the burden of workload for AMD screening might be significantly reduced in the future and implementation of screening protocols based on AI made easily accessible”.

• My minor remark:

a) Please mention preventive laser treatments for dry AMD (micropulse laser, 2-RT laser). Their efficacy is questioned at the moment, but the research is going on. If they have proven really preventive, than early screening would be justified.

Our response: Thank you for the comment. The manuscript has been updated with the following information:

“In the future, developing treatments to prevent AMD progression may improve the prognosis of the disease. Research is currently underway on preventive laser treatments for dry AMD, such as micropulse and nanosecond laser techniques [33–35]. The Early Stages of Age-Related Macular Degeneration (LEAD) study, for example, evaluated the safety of subthreshold nanosecond laser treatment in intermediate AMD and its efficacy for slowing progression to late AMD. The results found no significant reduction in the overall rate of progression to late AMD compared with sham treatment in intermediate AMD patients. However, post hoc analyses revealed a potential beneficial effect of subthreshold nanosecond laser (SNL) treatment in eyes without reticular pseudodrusen (RPD) at baseline, and that SNL treatment may increase the rate of progression to late AMD in eyes with RPD at baseline. As such, further studies are needed before recommendations can be made[36].”

Additional comment: We would like to inform you that Figure 1 (Study selection flowchart) was updated because the number of articles assessed for eligibility and the number of full articles excluded were revised.

---

## [Editor Report · Decision Letter 2]

2 Nov 2023

Clinical effectiveness of screening for age-related macular degeneration: a systematic review

PONE-D-23-03440R2

Dear Dr. Gomes,

We’re pleased to inform you that your manuscript has been judged scientifically suitable for publication and will be formally accepted for publication once it meets all outstanding technical requirements.

Kind regards,

Andrzej Grzybowski

Academic Editor

PLOS ONE
---

## [Editor Report · Acceptance letter]

7 Nov 2023

PONE-D-23-03440R2 

Clinical effectiveness of screening for age-related macular degeneration: a systematic review 

Dear Dr. Gomes:

I'm pleased to inform you that your manuscript has been deemed suitable for publication in PLOS ONE. Congratulations! Your manuscript is now with our production department. 

Kind regards, 

on behalf of

Dr. Andrzej Grzybowski 

Academic Editor

PLOS ONE